# Dietary-Polysaccharide-Modified Fish-Oil-Based Double Emulsion as a Functional Colloidal Formulation for Oral Drug Delivery

**DOI:** 10.3390/pharmaceutics14122844

**Published:** 2022-12-19

**Authors:** Shuzhen Li, Wanqiong Li, Xin Yang, Yanfeng Gao, Guanyu Chen

**Affiliations:** School of Pharmaceutical Sciences (Shenzhen), Shenzhen Campus of Sun Yat-sen University, Shenzhen 518107, China

**Keywords:** dietary polysaccharide, water-in-oil-in-water emulsion, colloidal delivery system, immune modulator, PD-L1 blocking peptide, oral drug delivery

## Abstract

Oral delivery is the most convenient drug administration route. However, oral delivery of peptides is extremely challenging due to the physical and chemical barriers within the gastrointestinal tract. Polysaccharides are often utilized as polymeric biomaterials in drug delivery. Among these, dietary polysaccharides extracted from okra, yam, and spirulina have been reported to stimulate innate immunity with well-known nutritional benefits. In this study, we developed a dietary-polysaccharide-modified fish-oil-based emulsion for oral co-delivery of a hydrophilic PD-L1 blocking peptide and the hydrophobic small molecule simvastatin. The optimal emulsion was nano-sized and exhibited a negative surface charge, high drug encapsulation efficiency of over 97%, low viscosity, and sustained drug release manner. The formulation could significantly increase the uptake of peptides by intestinal Caco-2 cells, which demonstrated the great potential of the formulation for promoting the oral absorption of peptides. Additionally, these dietary polysaccharides could promote dendritic cell maturation and cytokine expression in macrophages, demonstrating that these nutraceutical polysaccharides had dual roles of functioning as promising colloidal delivery systems and as potential immune modulators or adjuvants. Thus, this food-based colloidal delivery system shows promise for the oral delivery of peptide drugs and lays a great platform for future applications in immunotherapy.

## 1. Introduction

Oral drug delivery is the most conventional and convenient administration route for the treatment of various diseases, as it alleviates the pain caused by injections with desirable cost-effectiveness and shows a good safety profile [1]. However, the physical and chemical barriers of the gastrointestinal tract (GIT) drastically limit the oral absorption of drugs—particularly peptide drugs, which are easily degraded by the strong acids from the stomach and the proteolysis enzymes within the GIT [2]. Therefore, to overcome the physical and biochemical barriers in the GIT, the design of a suitable delivery system for promoting oral drug absorption is crucial.

Currently, emulsion technology has attracted much attention, as it has a low cost, involves a simple fabrication process with great stability, and can be easily scaled up in manufacturing; thus, it is widely applied in food, pharmaceutical, and nutraceutical products [3]. Double-phase emulsions, which consist of two types, oil-in-water-in-oil (O_1_/W/O_2_) and water-in-oil-in-water (W_1_/O/W_2_) are often used in the co-delivery of hydrophilic and hydrophobic bioactive compounds [4]. Although double-phase emulsions are kinetically stable, the outer layer of an emulsion may be degraded by high temperatures, oxidation, or strong acids or bases, and it may suffer from degradation into a single-phase emulsion during processing [5]. Therefore, finding an ideal material with appropriate properties for fabricating the outer layer of an emulsion is important.

At present, there has been increasing interest in the application of polysaccharides as polymeric biomaterials in drug delivery, since they are featured as biodegradable and biocompatible materials with controllable rigidity and functionality [6]. Additionally, these materials made from natural edible constituents show good safety profiles, and some dietary polysaccharides are even beneficial to patients in terms of general health and, thus, are widely consumed as nutraceutical supplements [7]. Moreover, some dietary polysaccharides have exhibited excellent emulsifying properties due to their higher chemical and colloidal stability; thus, the dietary-polysaccharide-modified emulsions have been introduced to provide appropriate stability of systems against the harsh environment in the GIT [8]. Some specific dietary polysaccharides show bio-adhesive properties, which allow for adhesion to the intestinal epithelial membrane and prolong drugs’ residence times in the GIT, thus promoting the drug absorption [9]. As functional food ingredients, okra fruit, yam, and spirulina have been noticed due to their great health benefits owing to their various biological activities, such as their antioxidant [10], anti-fatigue [11], immunomodulatory [12], anti-hyperglycemic [13], anti-hyperlipidemic, and anti-diabetic effects [14]. Indeed, okra, yam, and spirulina are all rich in bioactive polysaccharides, and they show great potential for being further explored as functional food ingredients for industrial applications in nutraceuticals and pharmaceutics [15].

The objective of this research was to establish a green delivery system for a dietary-polysaccharide-modified fish-oil-based W_1_/O/W_2_ emulsion to encapsulate a programmed death ligand-1 (PD-L1) blocking peptide and simvastatin. PD-L1 inhibitor is a major immune checkpoint inhibitor that is emerging as a front-line treatment for various types of cancer [16]. Our research group previously developed a PD-L1 blocking peptide, OPBP-1, with the sequence of Gqsehhmrvysf-NH_2_ (lower-case letters represent D-amino acids) and proved its excellent anti-tumor efficacy; thus, it was selected as a model peptide drug and encapsulated in the internal aqueous phase of a double emulsion [17]. Simvastatin is an oil-soluble drug for lowering lipids, and it is commonly used to treat cholesterol and heart problems. Additionally, it was reported that statins combined with PD-1/PD-L1 blockade therapy had excellent synergistic anti-tumor effects [18]. Thus, we designed the simultaneous delivery of the PD-L1 blocking peptide and the repurposed drug simvastatin through W_1_/O/W_2_ double emulsions. In this study, the double emulsion was characterized, which included the droplet size and size distribution, zeta potential, encapsulation efficiency, morphology, viscosity, and drug release. In addition, the drug uptake in Caco-2 cells was also determined. Moreover, to better understand the role of the polysaccharide-modified double-phase emulsion, the immune-modulating effect of the dietary polysaccharides was also investigated for the first time for the application of an emulsion in oral delivery. Thereby, this research has developed a green, easily fabricated delivery system for the oral delivery of a peptide drug and a small-molecule drug with the aim of enhancing the drugs’ oral bioavailability and of demonstrating that dietary polysaccharides have great potential for use as carrier systems and immune-modulating agents.

## 2. Materials and Methods

### 2.1. Materials, Cell Lines, and Mice

Fish oil (18% EPA, 20% omega-3) was purchased from Shanghai Yien Chemical Technology Co., Ltd. (Shanghai, China). Lecithin was purchased from Beijing Solarbio Technology Co., Ltd. (Beijing, China). Okra polysaccharide was purchased from Rundekang Biotechnology Co., Ltd. (Taipei City, Taiwan). Yam polysaccharide and spirulina polysaccharide were purchased from Mixianer Biotechnology Co., Ltd. (Xi’an, China). Glycerin was purchased from Shanghai Aladdin Biochemical Co., Ltd. (Shanghai, China). All other reagents were of analytical grade. Murine mononuclear macrophages (RAW 264.7 cells) and the human colon adenocarcinoma cell line (Caco-2 cells) were conserved in our laboratory. RAW 264.7 cells and Caco-2 cells were cultured in DMEM (Gibco, Grand Island, NE, USA). C57BL/6 female mice aged three to five weeks were maintained in a specific pathogen-free facility. Animal experimental procedures were carried out while following the national and institutional guidelines and were approved by the Ethics Committee of the Sun Yat-sen University.

### 2.2. Peptide Synthesis and Fluorescein Labeling

The standard solid-phase Fmoc synthesis method was used for peptide synthesis. Rink resin was used as a polymeric carrier. First, the resin was swelled and thoroughly washed. A hexahydropyridine/dimethyl formamide (DMF) mixture (1:4 or 1:1) was used to remove the protecting group of Fmoc from amino acids. After washing the resin with DMF, dichloromethane (DCM), the first amino acid, was added and attached to the resin by using a peptide condensation reagent, HCTU, and DIEA. The ninhydrin method was used to ensure the success of Fmoc removal and amino acid attachment. This process was repeated to obtain the initial peptide chain. Finally, the peptide was released from the resin by using trifluoroacetic acid, precipitated with ether, and centrifuged to obtain the crude peptide. Reverse-phase high-performance liquid chromatography (RP-HPLC) was used for peptide purification. Electrospray ionization mass spectrometry (ESI-MS) was used to verify the molecular weight of the peptide.

Fluorescein labeling of the OPBP-1 peptide was performed according to the literature [19]. The peptide sequence was FITC-PEG4-Gqsehhmrvysf-NH_2_, in which PEG4 was used as a linker between FITC and the PD-L1 peptide. The synthesis of PEG4-Gqsehhmrvysf-NH_2_ was performed by using the standard solid-phase Fmoc synthesis method, as mentioned above. After deprotection, excess FITC was dissolved in 4 mL of DMF and added into the reaction tube with 800 μL DIEA, shaken, and allowed to react overnight in the dark. Finally, the processes of peptide release from the resin, precipitation, purification, and identification were carried out.

### 2.3. Preparation of the W_1_/O/W_2_ Double Emulsion

The proportions of the inner aqueous phase (W_1_), the oil phase (O), and the external aqueous phase (W_2_) were 1:3:12. Lecithin (1.5% *w/v*) was dissolved in fish oil to obtain the oil phase. Deionized water was added to the oil phase dropwise and stirred for 10 min, followed by a homogenization process for 3 min by using ultrasonic processor. The primary W_1_/O emulsion was obtained. Different polysaccharide solutions (okra, yam, and spirulina polysaccharide) with different concentrations (1.25, 2.5, 5, 10, and 20 mg/mL) were prepared. Glycerin (8% *v/v*) was added to each solution to form an outer aqueous phase. The primary W_1_/O emulsion was added to the outer aqueous phase, and the dispersions were homogenized for 3 min to obtain the final W_1_/O/W_2_ double emulsions.

### 2.4. Stability Study of the W_1_/O/W_2_ Emulsions

We selected 3 dietary polysaccharides that had been reported to stimulate innate immunity: okra, yam, and spirulina polysaccharide. The polysaccharide solutions acted as the external aqueous phase of the emulsions. In order to explore the optimal polysaccharide concentration to form the most stable emulsion, 5 concentrations of 1.25, 2.5, 5, 10, and 20 mg/mL were investigated for each polysaccharide. Fifteen fresh emulsions were transferred into Eppendorf (EP) tubes and stored at 25 °C (room temperature) and 37 °C (body temperature) for 21 days. The stratification was observed, and 9 emulsions with poor stabilities were eliminated. As for the forced thermal stability study [20], 6 residual emulsions were placed in a warm-water bath at 45, 65, and 85 °C, respectively. The stratification was observed to eliminate unstable emulsions. Finally, 3 optimal polysaccharide-modified emulsions with the greatest stability were obtained, with one selected from each polysaccharide category.

### 2.5. Morphology

Samples were placed on the glass slides and gently covered with a coverslip to prevent the emulsions from being destroyed. An optical microscope (TS2, Nikon, Japan) was employed to observe the morphology of the emulsions at room temperature. A digital camera was used to acquire images.

### 2.6. Size and Zeta Potential

A Mastersizer 2000 (Malvern Instruments Ltd., Malvern, UK) was employed to determine the droplet size in the W_1_/O/W_2_ emulsions. D(v, 90) was used to represent the average droplet size of the emulsions. The zeta potential [21] was determined with a Zetasizer (Nano ZS, Malvern, UK). Measurements were in triplicate. Polydispersity was expressed by the SPAN factor [22]. The equation was as follows:SPAN = [D(v, 90) − D(v, 10)]/D(v, 50) (1)

D(v, 10), D(v, 50), and D(v, 90) are the volume size diameters at 10%, 50%, and 90% of the cumulative volume, respectively.

### 2.7. Encapsulation Efficiency

The encapsulation efficiency (EE) of the drugs in the double emulsion was determined by measuring the concentrations of the drugs in the external aqueous phase (W_2_ phase) [23]. The emulsions were centrifuged at 21,532× *g* at 4 °C for 15 min, and they were separated into creamy layers (upper, W_1_/O emulsion) and a whey layer (bottom, W_2_ phase). The whey layer contained the unencapsulated drug and was collected from each centrifugal sample by using a syringe. This process was repeated 2–3 times in order to completely remove the creamy layers. The EE% of the W_1_/O/W_2_ emulsions was estimated by using the equation:EE (%) = (1 − m_free_/m_total_) × 100% (2)
where m_free_ represents the amount of the drug penetrating into the external water phase (μg), and m_total_ represents the total amount of the drug added to the emulsion (μg).

### 2.8. Rheological Analysis

The viscosity of the emulsions was assessed by using a rheometer (MCR 302, Anton Paar GmbH, Austria). Shear flow tests were carried out to study the viscosity with a shear rate ranging from 0.1 to 100 s^−1^ at 25 °C [24].

### 2.9. Drug Release Study

To evaluate the drug release profile, 5 mL of free drug solution or 5 mL of drug-loaded double emulsions were transferred into a dialysis bag (molecular weight cutoff: 8000–14,000 Da) and dialyzed against 100 mL of simulated intestinal fluid (enzyme-free) at 37 °C with a stirring speed of 200 rpm/min. Samples were collected at specific time intervals. The concentration of the released drug was determined with a fluorescence microplate (SpectraMax Id5, China). The accumulative release of the drug was calculated.

### 2.10. MTT Assay

Intestinal Caco-2 cells were seeded into 96-well plates one night in advance for cell adherence. The next day, the medium was discarded, and serum-free DMEM medium was added for cell starvation. After 8 h, the serum-free medium was discarded. W_1_/O/W_2_ emulsions were added at 10% (*v*/*v*), 5% (*v*/*v*), 2.5% (*v*/*v*), 1.25% (*v*/*v*), 0.625% (*v*/*v*), and 0.3125% (*v*/*v*) and incubated for 2, 4, 6, 12, and 24 h, respectively. MTT was added into each well. When the time point was reached, the medium was discarded, and DMSO was added to fully dissolve the purple crystals. The absorbance was read at the OD of 490 nm.

### 2.11. In Vitro Drug Uptake Study by Caco-2 Cells via LSCM

Caco-2 cells were seeded in laser confocal cell dishes one night in advance for cell adherence. A replacement with clean culture medium took place the next day. A total of 100 μL of 0.2 mg/mL free FITC-modified peptide or peptide-loaded double emulsion was added and incubated with the cells for 2 h at 37 °C Then, the dishes were rinsed with phosphate-buffered saline (PBS) to remove residual FITC-modified peptides. Hoechst 33,258 was added and incubated for 30 min at 37 °C for nuclear staining. PBS was used to remove residual Hoechst. Laser-scanning confocal microscopy (LSCM) was employed to observe the peptide uptake by Caco-2 cells.

### 2.12. In Vitro Drug Uptake Study by Caco-2 Cells via Flow Cytometry

Briefly, Caco-2 cells were seeded into 48-well plate one night in advance for cell adherence. A replacement with fresh culture medium took place the next day. A total of 100 μL of 0.05 mg/mL free FITC-modified peptide or peptide-loaded double emulsion was added and incubated with the cells for 2 h at 37 °C. Then, the cells were collected and fixed, and the fluorescence intensity was measured via flow cytometry.

### 2.13. Effects of Polysaccharides on Dendritic Cell Maturation

This study was conducted by using bone-marrow-derived dendritic cells (BMDCs) from mice [25]. Briefly, the C57BL/6 mice were sacrificed, soaked in alcohol for 5 min, and then transferred to a sterile environment. The thigh bone was cut with scissors, and a syringe was used to flush out the bone marrow cells from the bone cavity. After filtration and centrifugation, the cell pellet was resuspended with ACK lysis buffer to lyse the red blood cells, washed with PBS twice, and cultured in a sterile culture dish with RPMI-1640 medium. GM-CSF and IL-4 were added into the medium for final concentrations of 20 and 10 ng/mL, respectively. Half of the medium was replaced daily from the third day while keeping the concentrations of GM-CSF and IL-4 unchanged. On the 6th day of induction, 2 × 10^5^ cells/well were cultured, and 5 groups were set up: negative control of PBS, positive control of 1 μg/mL lipopolysaccharide (LPS), and 50 μg/mL of okra polysaccharide, yam polysaccharide, and spirulina polysaccharide. After 24 h of incubation, the cells were collected, stained with anti-CD11c-APC, anti-CD80-PerCP-efluor^TM^ 710, and anti-CD86-PE, and analyzed via flow cytometry.

### 2.14. Effects of Polysaccharides on Macrophages

Murine macrophages (RAW 264.7 cells) were cultured in a 24-well plate at a density of 2 × 10^5^ cells/well, and the corresponding substances were added after starvation for 8 h. A total of 5 groups were set up: negative control of PBS, positive control of 1 μg/mL LPS, and 50 μg/mL of okra polysaccharide, yam polysaccharide, and spirulina polysaccharide. Over 24 h of incubation, the cells were lysed to extract RNA and reverse transcribed into cDNA, and the mRNA levels of IL-6 and TNF-α were detected via qPCR.

### 2.15. Statistical Analysis

The statistical analysis was conducted with a one-tailed Student *t*-test for the differences between two groups. All data are shown as the mean ± SD. * *p* < 0.05, ** *p* < 0.01, and *** *p* < 0.001.

## 3. Results

### 3.1. Optimization of Polysaccharide-Modified Emulsions

As shown in Figure 1A, among the okra polysaccharide emulsions, the 5, 10, and 20 mg/mL polysaccharide emulsions at 37 °C and the 10 and 20 mg/mL polysaccharide emulsions at room temperature appeared to undergo stratification. Among the yam polysaccharide emulsions, the 1.25 mg/mL polysaccharide emulsion at 37 °C and the 20 mg/mL polysaccharide emulsion at room temperature appeared to undergo stratification. Among the spirulina polysaccharide emulsions, the 2.5, 5, 10, and 20 mg/mL polysaccharide emulsions at 37 °C and the 10 and 20 mg/mL polysaccharide emulsions at room temperature appeared to undergo stratification. As a result, the relatively stable emulsions were the 1.25 and 2.5 mg/mL okra polysaccharide emulsions, the 2.5, 5, and 10 mg/mL yam polysaccharide emulsions, and the 1.25 mg/mL spirulina polysaccharide emulsion.

In the forced thermal stability studies at 45, 65, and 85 °C, the stratification of the emulsions was observed. As shown in Figure 1B, at 85 °C, the 2.5 mg/mL okra polysaccharide emulsion and the 5 and 10 mg/mL yam polysaccharide emulsions appeared to undergo stratification over 2 h. According to the 21-day storage stability and forced thermal stability study, the 1.25 mg/mL okra polysaccharide emulsion, 2.5 mg/mL yam polysaccharide emulsion, and 1.25 mg/mL spirulina polysaccharide emulsion remained stable; thus, they were selected for further studies.

Figure 1C shows the morphologies of the three optimal polysaccharide emulsions. The double-phase emulsion droplets appeared to have a round shape and the narrow size distributions without obvious aggregation, indicating the high uniformity of the droplets.

### 3.2. Droplet Size, Zeta Potential, SPAN Values, and Encapsulation Efficiency

The droplet sizes and zeta potentials of the emulsions were measured. As shown in Table 1, the droplet sizes of the three optimal polysaccharide emulsions were similar, ranging from 800 to 900 nm. They all showed a negative surface charge, ranging from −55 to −70 mV. SPAN is a measure of the width of the droplet size distribution. Generally speaking, the smaller the SPAN is, the narrower the droplet size distribution will be, and the larger the SPAN is, the wider the droplet size distribution will be. The three polysaccharide emulsions were also similar in terms of SPAN, which ranged from 3.30 to 3.50. The peptide was freely soluble in water, while the solubility of simvastatin in fish oil was determined to be 15 mg/mL. The EEs of these three polysaccharide emulsions for both the peptide drug and simvastatin were all over 97% (Table 1). The high drug encapsulation efficiency is also one major advantageous characteristics of double-phase emulsions.

### 3.3. Drug Release Study

Figure 2 shows the drug release profiles of the free peptides and peptide-loaded polysaccharide emulsions in simulated intestinal fluid (SIF). Free peptides were rapidly dissolved and permeated over the dialysis membrane over 45 min, while the okra polysaccharide emulsion had a drug release of 23.98 ± 1.66%, the yam polysaccharide emulsion had a drug release of 34.11 ± 1.84%, and the spirulina polysaccharide emulsion had a drug release of 36.70% ± 1.17 of peptides over the same period. The sustained release profiles of the three polysaccharide-modified emulsions allowed the elongation of the drug half-life and elevation of the overall oral bioavailability of the drugs.

### 3.4. Rheological Analysis

A flow curve is a graphical representation of how a flowing material behaves as the shear rate increases or decreases. The shape of the flow curve can distinguish the type of fluid. As shown in Figure 3, the viscosities of the three polysaccharide-modified emulsions did not increase or decrease with the increase in the shear rate from 0.1 to 100 s^−1^, which was consistent with the characteristics of a Newtonian fluid. This result reflected the low viscosity of the emulsions, which could improve the mouthfeel during oral administration, reduce the adhesion to the esophageal mucosa, and be friendly to patients with dysphagia or children and the elderly.

### 3.5. MTT Assay

As shown in Figure 4, none of the three polysaccharide-modified emulsions affected the growth of intestinal Caco-2 cells at 2, 4, and 6 h, demonstrating the good biosafety and biocompatibility of the polysaccharide-modified emulsions with the intestinal cells in vitro. Although it was observed that the growth of Caco-2 cells was slightly affected by the emulsions at 12 and 24 h, this could be negligible, since it was not expected that the oral emulsion would have such a long retention within the gastrointestinal tract in general.

### 3.6. In Vitro Uptake Study

As shown in Figure 5, all three polysaccharide-modified emulsions could significantly increase the uptake of the peptide drug by Caco-2 cells compared with the uptake of free peptides, which demonstrated the ability of the delivery system to promote the intestinal permeation of drug candidates.

Figure 6 shows the flow cytometry results, which indicate that all three polysaccharide-modified emulsions increased the FITC-labeled peptide uptake by Caco-2 cells compared to the uptake of free peptides. These drug uptake results that were analyzed via flow cytometry correspond to the uptake results of CLSM, which demonstrated that the polysaccharide-modified emulsions have good potential for promoting the oral absorption of peptide drugs.

### 3.7. Polysaccharides Promoted Dendritic Cell Maturation

As shown in Figure 7, okra and spirulina polysaccharides significantly increased the expression of CD80, CD86, and CD40 on the surface of the BMDCs, indicating that these two polysaccharides significantly promoted the maturation of BMDCs. Although yam polysaccharide did not significantly increase the levels of CD80, CD86, and CD40, the indexes also showed a slight upward trend. Mature BMDCs have a strong antigen-presenting ability and could effectively activate naive T cells to prime an immune response [26].

### 3.8. Polysaccharides Promoted Cytokine Expression in Macrophages

The effects of the polysaccharides on macrophages were also explored. Okra, yam, and spirulina polysaccharides significantly increased the IL-6 mRNA levels in RAW 264.7 cells. Okra and yam polysaccharides also increased the TNF-α mRNA levels (Figure 8). IL-6 can induce T cell proliferation and differentiation, thus participating in the immune response [27]. TNF-α can activate neutrophils and lymphocytes, regulate other tissues’ metabolic activities, and promote the release of other cytokines [28]. The results indicated that okra, yam, and spirulina polysaccharides could activate innate immunity, and they can act as immune-modulating agents in delivery systems.

## 4. Discussion

This study constructed a green oral delivery system based on dietary food. Fish oil is a well-known nutrient that reduces the risk of cardiovascular diseases and diabetes, relieves arthritis, depression, and anxiety, and has immunomodulatory and anti-tumor effects [29]. Dietary polysaccharides play a biological role in anti-tumor, anti-inflammation, anti-virus, hypoglycemic, antioxidant, anti-coagulation, and immune promotion activities, in addition to other aspects [30]. Three dietary-polysaccharide-based solutions were used as an outer aqueous phase to encapsulate a primary W_1_/O emulsion. The most stable formulations were obtained from a long-term storage stability study and a forced thermal stability study. The results showed that a lower concentration of the polysaccharide solution as the outer aqueous phase might be more conducive to the stability of the double emulsion. There might be a range of stability fluctuations between low-concentration polysaccharide solutions, which requires further investigation.

The optimal emulsions of the three polysaccharides had similar characteristics in terms of their nano-size, which is advantageous for oral absorption compared with micro-size droplets [31]. The emulsions all showed a negative surface charge, which was due to the presence of the negatively charged outer aqueous layer of polysaccharides [32]. The EEs of the peptide and simvastatin were all over 97% for the three optimal formulations. These results corresponds well with most of the findings in the literature, indicating that a high EE is a major advantageous characteristic of double-phase emulsions in general. Another significant advantage of W_1_/O/W_2_ double emulsions is their ability to simultaneously load a water-soluble drug and an oil-soluble drug within their internal aqueous phase and oil phase [33]. The emulsions were evaluated as Newtonian fluids, which indicated that the viscous stresses arising from their flow were linearly correlated with the local strain rate at every point [34]. The results showed the low viscosity of the emulsions. Less viscous emulsions are normally better for the mouthfeel in oral administration, and this is especially beneficial for patients with dysphagia, children, and the elderly [35].

The in vitro drug release from the W_1_/O/W_2_ emulsion was evaluated by using the technique of diffusion in a dialysis bag. The drug release profile was similar to that observed from double-phase emulsions reported in the literature, which is generally well correlated with pharmacokinetic results [36]. Irrespective of the nature of the sink solution, the drug release from the double-phase emulsions remained slow and incomplete as compared to that of a plain drug solution. This phenomenon was attributed to the presence of the oily internal phase, which led to a decrease in the aqueous drug gradient for membrane permeation, rendering the diffusion through the dialysis membrane the rate-limiting step in the overall kinetic process [37].

One of the main purposes of this study was to maximize the oral bioavailability of PD-L1 blocking peptides and improve the anti-tumor effects through the aid of carriers. In addition, studies have shown that PD-1/PD-L1 blockade therapy combined with simvastatin has better anti-tumor effects. Therefore, we chose simvastatin as a lipophilic drug that was loaded into the oil phase. The optimal polysaccharide-modified emulsions had high drug encapsulation efficiencies, which were very similar to those reported in the literature [38]. Additionally, the emulsions had a sustained manner of drug release, which is beneficial for the retention and absorption of drugs in the GIT, thus increasing the oral bioavailability and prolonging the half-life. As expected, the in vitro uptake experiments in Caco-2 cells demonstrated that all three polysaccharide-modified emulsions could significantly increase the uptake of peptides by Caco-2 cells compared to the uptake of free peptides, indicating the great potential of the delivery system to promote oral drug absorption and elevate the overall oral bioavailability of peptide drugs.

Finally, we evaluated the effects of okra, yam, and spirulina polysaccharides on the two major immune cells, dendritic cells and macrophages. More specifically, we investigated whether these polysaccharides could promote dendritic cell maturation and enhance cytokine expression in macrophages. The results indicated that okra and spirulina polysaccharides significantly increased the expression of CD80, CD86, and CD40 on the surface of BMDCs, indicating that these polysaccharides promoted dendritic cell maturation, which is conducive for antigen presentation and activation of naive T cells [39]. In addition, okra and yam polysaccharides also showed significant stimulation in RAW 264.7 cells to express the pro-inflammatory cytokines IL-6 and TNF-α. Therefore, this study demonstrated that these dietary polysaccharides showed the dual functions of forming promising carrier systems and being potential immune modulators. Additionally, it indicated that these dietary polysaccharides could activate natural immunity and might have a synergistic anti-tumor effect together with PD-L1 blocking peptides for immunotherapy.

## 5. Conclusions

In this study, dietary-polysaccharide-modified fish-oil-based double emulsions were successfully prepared through a simple emulsification method. The optimal emulsions were nano-sized and exhibited a negative surface charge, high drug encapsulation efficiency of over 97%, low viscosity, sustained drug release manner, and ability to improve oral drug absorption. In addition, these dietary polysaccharides could promote dendritic cell maturation and enhance cytokine expression in macrophages, which demonstrated that these dietary polysaccharides have dual roles of functioning as promising carrier systems and as potential immune modulators or adjuvants.

## Figures and Tables

**Figure 1 pharmaceutics-14-02844-f001:**
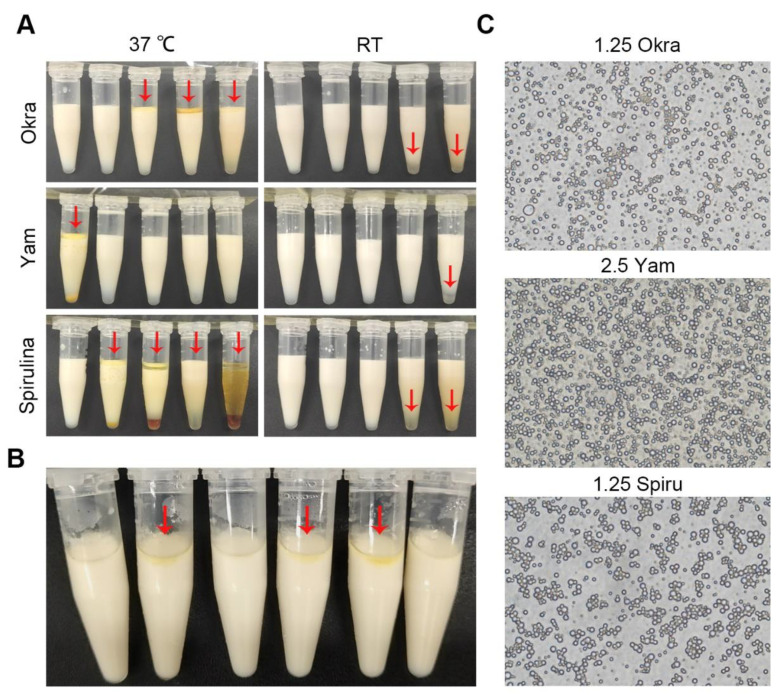
Stability study and morphology. (**A**) The 21-day storage stability. From left to right: 1.25, 2.5, 5, 10, and 20 mg/mL. Emulsions were stored at 37 °C and at room temperature for 21 days. (**B**) Forced thermal stability study. From left to right: 1.25 and 2.5 mg/mL okra polysaccharide, 2.5, 5, and 10 mg/mL yam polysaccharide, and 1.25 mg/mL spirulina polysaccharide emulsions. Here, the stratification of the emulsions at 85 °C over 2 h is shown. (**C**) An optical microscope was employed to observe the morphologies of the emulsions of 1.25 mg/mL okra polysaccharide, 2.5 mg/mL yam polysaccharide, and 1.25 mg/mL spirulina polysaccharide at room temperature.

**Figure 2 pharmaceutics-14-02844-f002:**
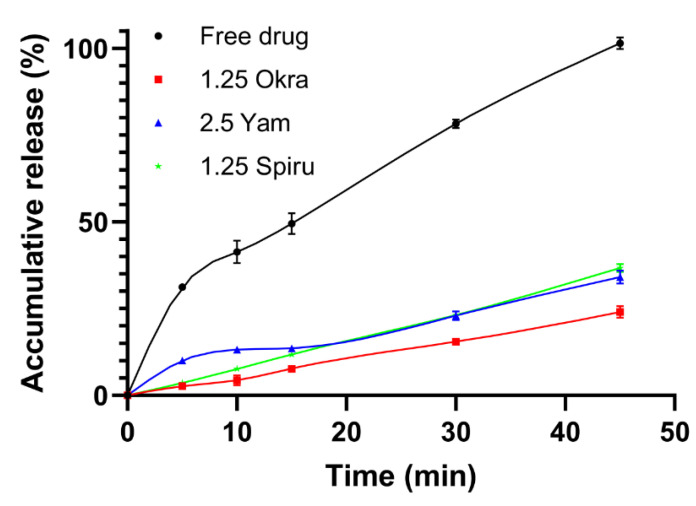
Drug release profile; 5 mL of free peptide solution or 5 mL of peptide-loaded double emulsion was transferred into a dialysis bag and dialyzed against 100 mL of artificial small intestine fluid (enzyme-free) at 37 °C. All data are reported as mean ± SD (n = 3).

**Figure 3 pharmaceutics-14-02844-f003:**
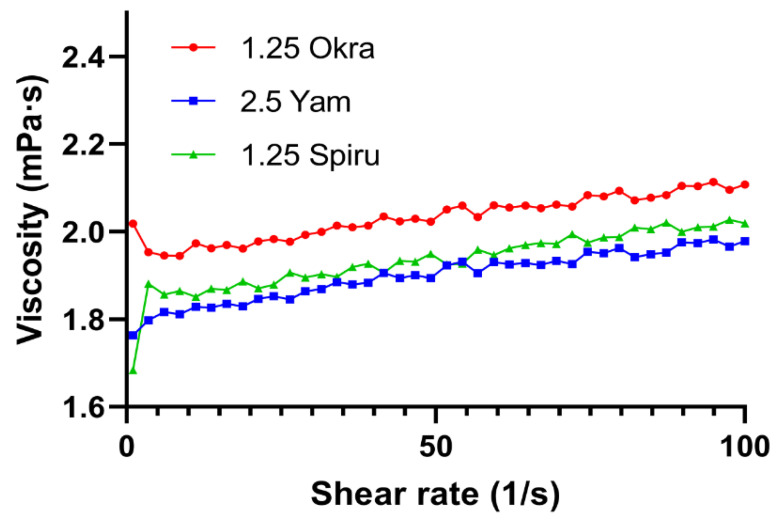
Rheological analysis of the three emulsions. Shear flow tests were carried out to study the viscosity with a shear rate ranging from 0.1 to 100 s^−1^ at 25 °C.

**Figure 4 pharmaceutics-14-02844-f004:**
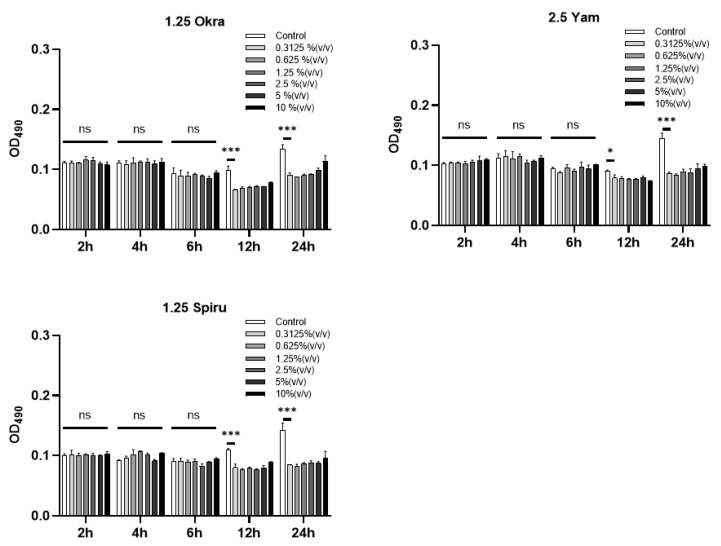
MTT assays of the three emulsions. Intestinal Caco-2 cells were cultured and six groups were set up: negative control and 10% (*v*/*v*), 5% (*v*/*v*), 2.5% (*v*/*v*), 1.25% (*v*/*v*), 0.625% (*v*/*v*), and 0.3125% (*v*/*v*) W_1_/O/W_2_ emulsions. The absorbance was read at OD = 490 nm. * *p* < 0.05; *** *p* < 0.001; ns, no significance.

**Figure 5 pharmaceutics-14-02844-f005:**
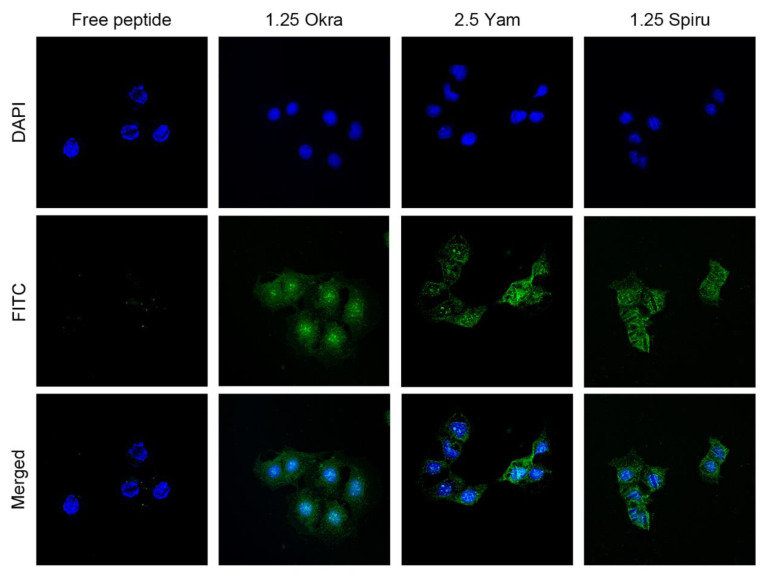
Peptide uptake by Caco-2 cells in vitro. A total of 100 μL of 0.2 mg/mL free FITC-modified peptide or peptide-loaded double emulsion was added and incubated for 2 h at 37 °C. LSCM was employed to observe the FITC-labeled peptide uptake by Caco-2 cells. Nuclei: blue; FITC-labeled peptides: green. The green-fluorescence-labeled peptides that appeared within the cytoplasm confirmed the sufficient cellular uptake of the peptide drugs.

**Figure 6 pharmaceutics-14-02844-f006:**
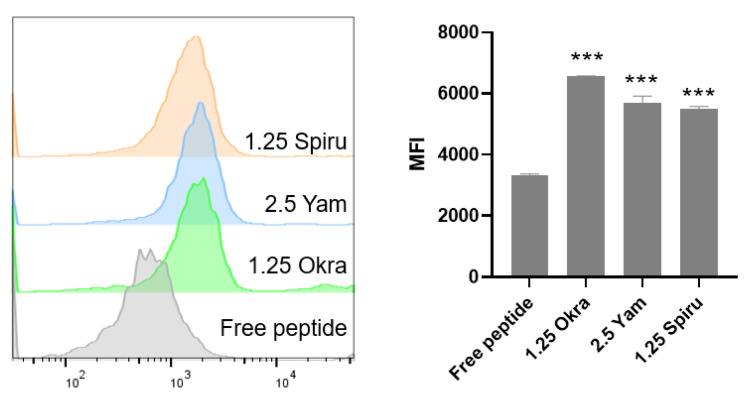
Peptide uptake by Caco-2 cells in vitro. A total of 100 μL of 0.05 mg/mL free FITC-modified peptide or peptide-loaded double emulsion was added and incubated for 2 h at 37 °C. Flow cytometry was employed to observe the FITC-labeled peptide uptake by Caco-2 cells. *** *p* < 0.001, ns, no significance.

**Figure 7 pharmaceutics-14-02844-f007:**
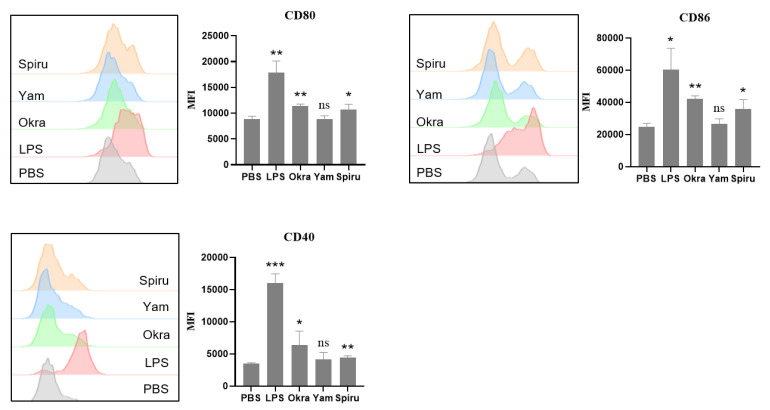
Effects of polysaccharides on BMDC maturation. Induced BMDCs were cultured, and five groups were set up: negative control of PBS, positive control of 1 μg/mL LPS, and 50 μg/mL of okra polysaccharide, yam polysaccharide, and spirulina polysaccharide. After 24 h of incubation, cells were collected, stained with anti-CD11c-APC, anti-CD80-PerCP-efluor^TM^ 710, and anti-CD86-PE, and assessed via flow cytometry. * *p* < 0.05; ** *p* < 0.01; *** *p* < 0.001; ns, no significance.

**Figure 8 pharmaceutics-14-02844-f008:**
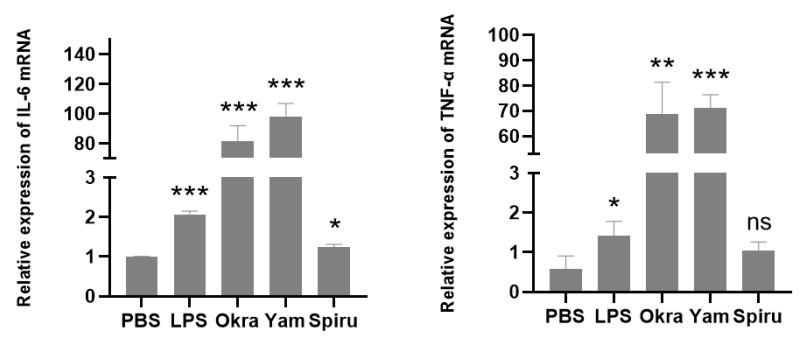
Effects of polysaccharides on RAW 264.7 cells. The RAW 264.7 cells were cultured, and the corresponding substances were added after starvation for 8 h. After incubation for 24 h, the cells were lysed to extract RNA and reverse transcribed into cDNA, and the mRNA levels of IL-6 and TNF-α were detected via qPCR. * *p* < 0.05; ** *p* < 0.01; *** *p* < 0.001; ns, no significance.

**Table 1 pharmaceutics-14-02844-t001:** The particle size, zeta potential, SPAN, and encapsulation efficiency of the emulsions.

Emulsion	Particle Size (nm)	Zeta Potential (mV)	Polydispersity (SPAN Value)	EE of Peptide (%)	EE of Simvastatin (%)
1.25 Okra	824.67 ± 1.53	−69.43 ± 1.37	3.34 ± 0.01	97.84 ± 0.07	98.16 ± 0.04
2.5 Yam	872.67 ± 2.52	−55.53 ± 1.22	3.44 ± 0.01	98.20 ± 0.09	98.30 ± 0.07
1.25 Spiru	800.00 ± 1.73	−61.37 ± 1.78	3.32 ± 0.01	97.69 ± 0.05	97.60 ± 0.06

Data are shown as mean ± standard deviation (n = 3).

## Data Availability

All the data is contained within the article.

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
