# Peer review of "Dietary-Polysaccharide-Modified Fish-Oil-Based Double Emulsion as a Functional Colloidal Formulation for Oral Drug Delivery"

_pharmaceutics, 2022, doi:10.3390/pharmaceutics14122844_

Round 1

Reviewer 1 Report

Dear Authors,
your manuscript  "Dietary polysaccharide modified fish oil-based double emulsion as a functional colloidal formulation for oral drug delivery " is very interesting but I believe that further experiments on the safety of your chosen nanoemulsions are needed. I would like to ask you to schedule experiments using normal human cells to evaluate the biocompatibility of your nanoemulsions.
Furthermore, you should explain to me how you have established the concentrations of the various cargoes in the nanoemulsion.

Furthermore, you should provide the quantitative uptake data by performing experiments in flow cytometry.

Best regards

Reviewer 2 Report

In this manuscript by Li S. et al., the authors developed a polysaccharide-modified fish oil-based emulsion for oral co-delivery of a PD-L1 blocking peptide. The manuscript addresses the complex challenge of peptide delivery due to their high susceptibility to proteases. The manuscript is well-written with targeted studies and desired results.

I have just some minor points:

-       Introduction (page 2): Please, add the PDL-1 peptide sequence in the line: “our research group previously developed..”

-       In Materials and methods, the peptide synthesis and fluorescein labeling should be described.

-       Please, the resolution of figures 2 and 3 should be improved.

Round 2

Reviewer 1 Report

Dear Authors, 

your manuscript " Dietary polysaccharide modified fish oil-based double emulsion as a functional colloidal formulation for oral drug delivery" can be accept in present form.

Best regards